# Adherent-Invasive *E. coli*: Update on the Lifestyle of a Troublemaker in Crohn’s Disease

**DOI:** 10.3390/ijms21103734

**Published:** 2020-05-25

**Authors:** Mélissa Chervy, Nicolas Barnich, Jérémy Denizot

**Affiliations:** 1Université Clermont Auvergne, Inserm U1071, USC-INRAE 2018, Microbes, Intestin, Inflammation et Susceptibilité de l’Hôte (M2iSH), 63001 Clermont-Ferrand, France; melissa.chervy@uca.fr (M.C.); nicolas.barnich@uca.fr (N.B.); 2Institut Universitaire de Technologie, Génie Biologique, 63172 Aubière, France

**Keywords:** Crohn’s disease, adherent-invasive *Escherichia coli*, AIEC-host interaction, virulence gene, intestinal environment, AIEC-targeting therapy, microbiota

## Abstract

Besides genetic polymorphisms and environmental factors, the intestinal microbiota is an important factor in the etiology of Crohn’s disease (CD). Among microbiota alterations, a particular pathotype of *Escherichia coli* involved in the pathogenesis of CD abnormally colonizes the intestinal mucosa of patients: the adherent-invasive *Escherichia coli* (AIEC) pathobiont bacteria, which have the abilities to adhere to and to invade intestinal epithelial cells (IECs), as well as to survive and replicate within macrophages. AIEC have been the subject of many studies in recent years to unveil some genes linked to AIEC virulence and to understand the impact of AIEC infection on the gut and consequently their involvement in CD. In this review, we describe the lifestyle of AIEC bacteria within the intestine, from the interaction with intestinal epithelial and immune cells with an emphasis on environmental and genetic factors favoring their implantation, to their lifestyle in the intestinal lumen. Finally, we discuss AIEC-targeting strategies such as the use of FimH antagonists, bacteriophages, or antibiotics, which could constitute therapeutic options to prevent and limit AIEC colonization in CD patients.

## 1. Introduction

Crohn’s disease (CD) and ulcerative colitis (UC) are inflammatory bowel disease (IBD) characterized by chronic inflammatory disorders of the gastrointestinal (GI) tract. CD, first described in 1932 by the doctor Burill B. Crohn [1], is an inflammatory disease of the intestinal mucosa developing preferentially in the young adult and evolving in a relapsing and remitting manner. It can affect all the segments of the GI tract with preferential localization in the terminal ileum and the colon. To date, no curative treatment exists for CD patients, only symptomatic treatments are proposed to limit the frequency and the intensity of the inflammatory flare [2].

The etiology of CD is multifactorial, resulting from the interplay between genetic susceptibility, environmental factors, and abnormal intestinal microbiota composition. Several genetic polymorphisms increase the risk of developing CD. As examples, three polymorphisms in the gene *NOD2*, involved in the intracellular recognition of bacterial muramyl dipeptide (MDP), have been highly associated to a higher risk to develop CD in many cohorts of patients [3,4]. Genes involved in the autophagy pathway, important for the elimination of intracellular bacteria, such as *ATG16L1* and *IRGM*, have also been found mutated in CD patients [5,6,7]. CD is, however, not a genetic disease as many studies observed discordances in monozygotic twins [8,9,10], indicating the involvement of other factors in its etiology. Talking in regards with the environmental factors, smoking has been identified as a risk factor for the development of CD, and consumption of Western diet (low-fiber diet enriched in total fat and sugar) and dietary emulsifiers (present in processed foods) are together associated with a higher susceptibility to develop CD [11,12,13,14]. The role of microbial composition of the intestinal microbiota has also been largely studied for its implication in the etiology of CD. Dysbiosis has been observed in CD patients characterized by a reduction of the global microbial diversity and, more specifically, a decrease in beneficial bacteria such as those of the *Firmicutes* phylum and an increase of the *Bacteroidetes* and *Proteobacteria* phyla [15,16,17,18,19,20]. Enterobacteria, members of the *Proteobacteria* phylum, can represent up to 100% of the aeroanaerobic microbiota associated to the ileal mucosa in CD, and particular adherent strains of *Escherichia coli* have been identified in the inflamed ileal and colonic mucosa of CD patients [21,22,23,24,25,26,27]. Both abilities of these strains to adhere to and to invade intestinal epithelial cells (IECs) as to survive and to replicate within macrophages led to the definition of a new pathogenic group: the adherent-invasive *E. coli* (AIEC) [21,22,28]. Preclinical models of CD suggest an important role of these bacteria in the induction and/or maintenance of intestinal inflammation in CD patients [29,30,31]. In this review, we describe the lifestyle of AIEC bacteria within the intestine, from the interaction with intestinal epithelial and immune cells with an emphasis on environmental and genetic factors favoring their implantation, to their lifestyle in the intestinal lumen. Based on the recent literature, this review will also discuss the new therapeutic strategies which could be developed to limit AIEC overgrowth in CD patients and ongoing clinical trials targeting AIEC bacteria.

## 2. The Challenging Identification of AIEC Bacteria in CD Patients

The prevalence of AIEC bacteria in the mucosa ranges from 21% to 62% in CD patients versus 0% to 19% in healthy controls, AIEC being found more frequently associated to the ileum than to the colon [22,24,32]. AIEC bacteria are also detected in healthy controls, at a lower rate than in CD patients, without inducing CD-specific symptoms. These bacteria occur not to be strict pathogens, as defined by the Koch’s postulate, but to be pathobiont bacteria which take advantage of host genetic alterations and/or take advantage of a specific environment to favor their implantation in ileal mucosa to induce intestinal inflammation in CD patients.

Despite some metabolic abilities (use of propanediol and iron uptake), different transcriptomic profiles of AIEC vs. commensal *E. coli*, and enrichment in specific polymorphisms in virulence genes, no specific gene for all AIEC strains has been detected, nor molecular markers common to all AIEC, making it difficult to identify AIEC-positive patients [33]. A study involving CD patients called “MOlecular BIomarkers and Adherent and Invasive *Escherichia coli* (AIEC) Detection Study in Crohn’s Disease Patients” (MOBIDIC) was performed to evaluate the relationship between noninvasive biomarkers and AIEC detection in intestinal biopsies. The objective was to develop a predictive algorithm of AIEC carriage, but the results of this trial have yet to be released (NCT02882841). In a recent study, Camprubí-Font et al. identified three AIEC-associated single nucleotide polymorphisms (SNP), and they elaborated from these SNP an algorithm able to predict AIEC phenotype with high accuracy (84%), showing that these SNP could be used to predict the carriage of AIEC [34]. However, it has recently been demonstrated that this prediction of AIEC phenotype was not reproducible with *E. coli* strains from different cohorts meaning that it is not a robust method to identify AIEC [35]. Hence, to date, the only way to identify AIEC strains is to evaluate the ability of mucosa-associated *E. coli* to invade IECs and to survive within macrophages in vitro.

Many research groups now focus their work to better understand how AIEC bacteria evolved from commensal *E. coli* strain, how AIEC bacteria interact with the complex microenvironment of the digestive tract, how they take advantage of specific environmental conditions, how they interact with IECs, and how they subvert host defense to colonize intestinal mucosa of CD patients. The common efforts should lead to get a better grasp of the physiology of these bacteria to propose new diagnostic approaches to identify CD patients colonized by AIEC bacteria easily and should lead to the development of new therapeutic approaches to eliminate AIEC bacteria.

## 3. Access to the Mucosa

### 3.1. Mucus Crossing

A protective mucus layer covers the surface of the intestinal epithelium and is composed of gel-forming glycoproteins called mucins, secreted by specialized goblet cells [36]. The mucus layer is central in the protection against many intestinal pathogens, such as *Shigella flexneri, Yersinia enterocolitica,* and *Salmonella enterica* [37,38]. Carbohydrate structures on mucins are targeted by pathogenic bacteria for their attachment to intestinal mucosa, avoiding their elimination through the natural transit [39]. In CD patients, the mucus layer is essentially continuous and comparable to healthy mucosa, although abnormal expression and glycosylation of the mucins have been noticed, which renders the mucus layer more penetrable by commensal and pathogenic bacteria [40,41,42,43,44]. Interestingly, a study using IECs lines revealed that AIEC strain LF82 (one of the AIEC reference strains isolated from CD patients) infection decreased the expression of MUC2 and MUC5A [45]. Pathogenic bacteria, such as Enteroaggregative *Escherichia coli* (EAEC), Enterotoxigenic *Escherichia coli* (ETEC), or *Shigella flexneri* possess mucinase activity allowing the degradation of mucins to facilitate their passage through the protective layer. A study by Gibold et al. revealed that AIEC bacteria strain LF82 exhibited a mucinolytic activity mediated by the bile salt- and mucin-mediated upregulation of *vat-AIEC* gene expression. The expression of this protease by AIEC bacteria conferred a better ability to colonize the digestive tract of genetically susceptible CEABAC10 mice overexpressing human CEACAMs molecules, mimicking CD susceptibility [29,46]. Their study also described the role of Vat-AIEC mucinase in the spread of AIEC in the mucus, by decreasing the viscosity of the mucus, and, as a consequence, in the access to epithelial cells (Figure 1). These data show that the mucinase Vat-AIEC contributes to the virulence of AIEC at the first step of infection as it favors the penetration of AIEC through the mucus layer and their adhesion to IECs.

### 3.2. Flagellum Regulation

AIEC bacteria flagella, which are filamentous organelles that allow locomotion, bacterial mobility, encoded by the *fliC* gene coding for FliC protein being the major component of flagellum, plays a central role in the adhesion process of the bacteria to IECs directly, *via* motility and indirectly, by maintaining the expression of type 1 pili [47,48]. Moreover, flagellum expression is directly involved in the induction of intestinal inflammatory response, through its binding to Toll-like receptor 5 (TLR5), an activator of the innate immunity (see below).

A recent study noteworthy identified the hypermotile phenotype as selected during evolution of AIEC under gastrointestinal conditions. The passage of an ancestor AIEC strain in the digestive tract of mice and the natural transmission between mice led to the selection of AIEC strains with a hypermotile phenotype and an increased fitness in vivo (Figure 1). Genome sequencing revealed that the hypermotility of selected strains was due to the integration of a mobile insertion sequence upstream of the master flagellar regulator, *flhDC*, which enhanced AIEC invasion and promoted the establishment of a mucosal niche [49].

Importantly, flagellin is also expressed by commensal bacteria without inducing uncontrolled inflammation suggesting specific regulation of flagellin expression in AIEC bacteria during the colonization process. A recent paper from Sevrin et al. uncovered an interesting regulation mechanism of *fliC* gene expression in AIEC bacteria. A different regulation of *fliC* gene expression was noticed in response to intestinal microenvironment components between AIEC bacteria and commensal *E. coli* HS strain. Indeed, a gene reporter system was used to reveal that *fliC* gene was upregulated in response to bile salts and mucus in AIEC bacteria but not in commensal strains. Thus, these specific features would then confer a selective advantage to AIEC pathobiont for the colonization of ileal mucosa of CD patients. To highlight the effect of *fliC* gene expression during intestinal colonization, the authors created an AIEC strain constitutively overexpressing *fliC*, through the deletion of the negative regulator *flgM* gene, and observed that this strain induced uncontrolled inflammatory response in the mice leading to its rapid clearance from the intestine, unlike the wild-type strain. From these facts, we would conclude that AIEC bacteria are able to regulate finely flagellum production to avoid overactivation of the immune system, allowing their long-term persistence in the gut [50].

### 3.3. Resistance to Antimicrobial Peptides

Paneth cells, found in the crypt of the small intestine, secrete antimicrobial peptides (defensins) such as α- and β-defensins, cathelicidins, RegIII family antimicrobial lectins, and lysozyme which diffuse, following a concentration gradient, within the mucus layer [51]. These defensins are involved in the maintenance of host–microorganisms homeostasis through the killing of bacteria [52]. In this context, AIEC bacteria must either take advantage of deficiencies in defensins secretion in CD patients or develop resistance to antimicrobial peptides (Figure 1). A paper from McPhee et al. identified 2 genes involved in the antibacterial peptides resistance in AIEC strain NRG857c isolated from a CD patient. They observed that the plasmid carried by this specific AIEC strain (pO83) comprises genomic island containing two genes involved in resistance to cationic antimicrobial peptides: *arlA* and *arlC*. *arlA* encodes a Mig-14 family protein implicated in defensin resistance, whereas *arlC* encodes an OmpT family outer membrane protease. The authors nicely confirmed the role of these genes in the resistance of AIEC strain NRG857c and other strains from CD patients to antimicrobial peptide using in vitro and in vivo approaches [53]. 

Thus, through diverse virulence factors, AIEC bacteria are able to cross the mucus layer and to regulate finely their flagella expression to reach the intestinal epithelium where they will use different strategies to favor their encroachment to the intestinal mucosa.

## 4. Interaction with the Intestinal Epithelium: Adhesion to IECs

### 4.1. Type 1 pili-CEACAM6 Interaction

Once in contact with IECs, AIEC bacteria use different strategies to interact with the cell and to invade it. One of the best-characterized interactions is the Carcinoembryonic antigen-related cell adhesion molecule 6 (CEACAM6)-FimH interaction. FimH is an adhesin located at the tip of type 1 pili highly expressed by AIEC bacteria and coregulated with flagella expression. Type 1 pili mediate bacterial adhesion to IECs and also play an important role in the ability of AIEC to invade IECs [54]. Some recently acquired mutations in *fimH* gene have been identified as conferring AIEC bacteria a significantly higher ability to adhere to IECs, strains harboring these mutations mostly belong to the B2 phylogroup, which regroups the most pathogenic *E. coli* strains. These data show that the important ability of AIEC to adhere to IECs could be the consequence of pathoadaptive mutations acquired during evolution and selected by gastrointestinal environment [55]. A work from Barnich et al. identified the highly mannosylated glycosylphosphatidylinositol (GPI)-anchored protein CEACAM6 as a receptor for AIEC bacteria. Indeed, the authors observed that mannose addition to the medium or *fimH* mutation in AIEC bacteria inhibited the adhesion of AIEC bacteria to human enterocytes from CD patients. Moreover, antibodies directed against CEACAM6 protein also prevented the interaction [56]. A more recent study better characterized CEACAM6-FimH interaction. Dumych et al. showed that oligomannose glycans exposed on early apoptotic cells are the binding targets of AIEC, which would mean that apoptotic cells could serve as potential entry points for bacteria into the epithelial cell layer before propagating literally in the intercellular spaces. The authors also identified two oligomannosidic glycans (Asn-197 and Asn-224) in CEACAM6 protein which could be potential receptors for the FimH adhesin [57]. 

These data obviously showed that AIEC bacteria bind to the CEACAM6 molecule and that this interaction depends on FimH adhesin expression and mannose residues on CEACAM6. Interestingly, in physiological conditions, CEACAM6 is not expressed by ileal epithelial cells but this protein has been found expressed at an important level in ileal mucosa of CD patients [56,58]. *In vitro,* AIEC bacterial infection of IECs led to the induction of *CEACAM6* gene expression, due to hypoxia-inducible factor-1 (HIF-1) stabilization, implying that these bacteria can favor their own colonization in CD patients by activating signaling pathways in the host cell [59,60]. 

### 4.2. ChiA-Chitinase 3-Like-1 Interaction

AIEC can also adhere to IECs through other host factors such as the Chitinase 3-like-1 (CHI3L1). CHI3L1, belonging to the glycohydrolase 18 family of chitinases, contains chitin-binding domain (CBD) and was reported to be upregulated during intestinal inflammation, predominately on IECs and lamina propria (LP) macrophages [61,62]. Interestingly, an acute colitis can be exacerbated *via* CHI3L1 overexpression favoring bacterial adhesion and internalization into IECs [61]. The bacterial *chiA* gene was identified in AIEC bacteria as involved in the interaction with IECs, and five conserved polymorphisms within the *chiA* gene were identified in pathogenic *E. coli* strains compared to nonpathogenic *E. coli* strains notifying that these polymorphisms may have been selected and could be associated to the virulence of AIEC bacteria. The interaction is mediated by the CBD of the AIEC chitinase ChiA that recognizes CHI3L1, and, more specifically, the *N*-glycosylation of asparagine 68 residue in mouse CHI3L1, this specific interaction promoting pathogenic effects of AIEC in mice with colitis [63].

### 4.3. AIEC-M-cells Interaction

Host-AIEC interaction can also occur through the follicle-associated epithelium (FAE) overlying Peyer’s Patches (PP), where M-cells are located, *via* Long Polar Fimbriae (LPF) expression by the bacteria. Indeed, a *lpf*-negative AIEC mutant was highly impaired in its ability to interact with mouse and human PP and to translocate across monolayers of M-cells (specialized cells of FAE), demonstrating that LPF, whose expression is dependent of bile salts concentration is a key factor for AIEC-PP interaction [64,65,66]. AIEC could also bind M-cells *via* the interaction between the FimH adhesin and the glycoprotein 2 (GP2), expressed on the apical membrane of M-cells promoting mucosal immune response to AIEC [67]. Deficiency of host GP2 altered transcytosis of type 1-piliated bacteria through M-cells and reduced the immune response in PP demonstrating that GP2 serves as a transcytotic receptor. In addition, colonic M cells express the highly glycosylated CEA and CEACAM1 molecules at their apical surface which could potentially play the role of receptors through a FimH-CEACAM interaction [68].

### 4.4. Invasion of IECs by AIEC Bacteria

Outer membrane vesicles (OMV), whose production is dependent on the *yfgl* gene, are involved in the invasive ability of AIEC LF82, by delivering bacterial effectors to host cells [69]. The major protein on the surface of OMV, OmpA, binds to the ER-localized stress response protein Gp96. Gp96 is overexpressed on the apical surface of ileal IECs in CD patients, and the OmpA-Gp96 interaction favors the fusion of OMV with IECs and promotes AIEC invasion through the delivery of proteins and different molecules within the host cell [70,71]. 

The pathogenic role of *ibeA* gene has been reported in many pathogenic bacteria. Of note, different strains of AIEC bacteria carry this gene (NRG857c, LF82, UM146, and KD-1) suggesting a potential role of this gene in the virulence of AIEC bacteria. A study revealed the role of *ibeA* gene in the interaction of AIEC with IECs. In vitro, an *ibeA*-deleted mutant AIEC strain was impaired in its ability to invade IECs, M-like cells, and to survive within macrophages, but was not impaired in its adhesion ability, supporting the fact that IbeA protein is involved in the AIEC invasion process. Finally, in vivo experiments revealed that *ibeA* was involved in the inflammatory response induced during intestinal colonization of AIEC [72]. 

In order to prevent AIEC entry in host cells, it seems necessary to identify new bacterial invasins and human proteins needed for the invasion process of IECs by AIEC. The genes coding for invasins could be identified using a transposon mutagenesis coupled with next-generation sequencing (Tn-seq) approach, which is based on the generation of a transposon insertion library and the identification of mutants unable to invade host cells. The lost mutants might harbor the transposon in either an essential gene for bacterial growth or in genes necessary for AIEC adhesion and/or invasion. Transposon-disrupted genes can be identified through next-generation sequencing of the flanking regions of the transposon insertion in recovered bacteria [73]. This strategy could be relevant to determine new virulence factors in AIEC bacteria not only for interaction with host cells but also for its fitness in vivo.

## 5. Consequences of AIEC Interaction with Epithelial Layer

### 5.1. Effects of AIEC on Epithelial Barrier Function

The epithelial cells from the intestine are interconnected by tight junctions (TJ) and adherent junction proteins, forming the apical junctional complex (AJC) to create a selectively permeable barrier necessary to protect the host against pathogenic bacteria and their toxins [74,75]. Deficient barrier function is frequently observed in CD patients and is predictive of relapses in CD [76]. The effect of AIEC infection on barrier function has been studied in different models. Several studies demonstrated in vitro that infection with AIEC decreased transepithelial resistance of an IECs monolayer and induced the delocalization of the tight junction adaptor protein ZO-1 and E-cadherin and the disorganization of F-actin. AIEC might then be involved in the defective barrier function observed in CD patients [77,78]. AIEC infection of CEABAC10 mice led to an increase in intestinal permeability, in a type-1 pili-dependent manner, and to the induction of pore-forming claudin-2 expression in IECs. Co-overexpression of claudin-2 and CEACAM6 was reported in quiescent and active phases CD patients, although the presence of AIEC bacteria associated to the mucosa has not been defined in this study [79]. 

Many factors have been suggested as favoring the ability of AIEC to colonize the intestinal epithelium and altering barrier function. Remarkably, the overexpression of *CEACAMs* genes in the mouse model of CD (CEABAC10 mouse model) was associated to an abnormally increased intestinal permeability, without signs of inflammation in the colonic mucosa, pointing that abnormal expression of *CEACAMs* genes, as notified in ileal mucosa of CD patients, could alter the integrity of the epithelial layer by itself. Importantly, the CEABAC10 mice are highly sensitive to dextran sulfate sodium (DSS)-induced colitis, confirming a barrier defect at a steady state upon CEACAMs overexpression [79]. High-fat diet and vitamin D deficiency have also been associated to altered barrier function. Of note, high-fat/high-sugar (HF/HS) diet favors AIEC colonization and vitamin D deficiency predisposes to AIEC-induced barrier dysfunction [80,81]. 

All these experimental data prove that AIEC bacteria can increase intestinal permeability in CD patients but also take advantage of already altered intestinal barrier by environmental factors or genetic susceptibility. 

### 5.2. Effects of AIEC on Host Glycosylation

Glycobiology, which consists in the study of functional roles of sugar in the cell, has emerged as potentially interesting to bring to the notice of the etiology of CD and to identify new therapeutic targets [82]. In a recent study, Sun et al. showed that level of O-Linked β-N-acetylglucosamine (O-GlcNAc) is increased in intestinal tissue of CD patients when compared to healthy controls [83]. Intestinal epithelial cells from mice and human cell lines infected with the AIEC reference strain LF82 presented a striking increase in the level of UDP-GlcNAc, a donor glucosamine for glycosylation, which was associated to an increase in O-GlcNac. As a consequence, O-GlcNac on NF-κB facilitated its nuclear translocation and the transactivation of NF-κB targeted inflammatory genes and limited autophagy-mediated elimination of intracellular AIEC bacteria. The authors managed to reverse the effect by using an inhibitor leading to depletion of O-GlcNAc in IECs and observed an overactivation of autophagy and anti-inflammatory effect in mice treated with the inhibitor, infected with AIEC bacteria, and treated with DSS. 

This study is preliminary in the field of glycobiology and AIEC bacteria as many other cellular pathways might be modulated by AIEC bacteria in a glycosylation-dependent manner, opening new research avenues for therapeutic targets for CD patients.

### 5.3. Induction of Fibrosis by AIEC

Tissue fibrosis is often detected in CD patients due to chronic inflammation and it leads to extensive local intestinal fibrosis and mechanical bowel obstruction that may require surgical resection and endoscopic dilation [84]. Small et al. demonstrated that chronic AIEC infection triggered intestinal inflammation and fibrosis, the factors leading to the establishment of fibrosis remaining unknown [85]. In a mouse model of IBD, the IL-10 KO mice model developing spontaneous colitis, germ-free mice infected with yersiniabactin-producing AIEC developed fibrosis. Infection with a Δ*fyuA* mutant, unable to import the Ybt siderophore, resulted in increased expression of profibrogenic genes, enhanced subepithelial invasion, and exacerbated fibrosis compared to infection with wild-type (WT) strain. In contrast, mice infected with the isogenic mutant Δ*irp1,* unable to synthesize Ybt, as well as with the Δ*fyuA*Δ*irp1* mutant presented reduced fibrosis incidence demonstrating that fibrosis development depends on the biosynthesis of Ybt, which creates a profibrotic environment when not associated with its receptor [86]. In a mouse model of AIEC infection followed by acute inflammatory trigger (DSS or *Salmonella* infection), AIEC bacteria stably colonized the inflamed gut, leading to massive deposits of collagen and consequently fibrosis, which was not observed in nonpathogenic *E. coli*-infected mice [87]. This work revealed that AIEC induce fibrosis through the activation of the IL-33-ST2 signaling, through the flagellin-induced expression of IL-33 receptor ST2. The binding of IL-33 to its receptor ST2 leads to the activation of NF-κB and MAPK pathways and to the production of pro-inflammatory cytokines that promote TGF-β and collagen deposition in the extracellular matrix [88,89]. These results show that AIEC can induce fibrosis upon chronic intestinal colonization and inflammation, *via* flagellin or the biosynthesis of siderophore.

### 5.4. Induction of Inflammation during AIEC Colonization

#### 5.4.1. Through the TLR Pathway

Once the AIEC–host interaction is established, host response sets up like induction of an inflammatory response characteristic of CD. Flagellum expression is directly involved in intestinal inflammatory response. Indeed, flagellin is a microbe-associated molecular pattern (MAMP) recognized by innate immune system receptors: the transmembrane TLR5 and the cytoplasmic receptor NLR family CARD domain-containing protein 4 (NLRC4) [30,90,91]. The pro-inflammatory role of flagellin has been confirmed in a mouse model of infection with AIEC bacteria where Carvalho et al. observed that AIEC strain LF82 infection increased expression of TLR5 and NLRC4 and aggravated DSS-induced colitis in mice characterized by severe histopathological damage and increased expression of *IL-1β* and *IL-6* in a flagellin expression-dependent manner [92]. 

Lipopolysaccharide (LPS) is also a MAMP that can activate TLR4 receptor and elicits intestinal inflammation. However, under healthy conditions, IECs down-regulate TLR4 expression to limit its activation and uncontrolled inflammatory response [93]. MicroRNAs (miRNAs), small noncoding RNA that post-transcriptionally regulate genes expression, are involved in various processes such as cellular proliferation/differentiation and immune response to pathogens. Misregulation of miRNAs has been associated with several disorders including IBD [94,95]. Guo et al. demonstrated in IL-10 KO mice that AIEC LF82 infection exacerbated colitis and, unlike non-infected mice, AIEC-infected mice presented decreased miRNA let7b, increased TLR4 expression, and increased secretion of pro-inflammatory cytokines (IL-6, IL-8, and tumor necrosis factor alpha (TNF-α)) in colonic epithelial cells. In vitro, in T84 cells, overexpression of TLR4 and consequently the release of pro-inflammatory cytokines was due to the reduced expression of let-7b caused by AIEC infection [96]. These effects were reversed in vivo and in vitro with the overexpression of let-7b. Furthermore, inhibition of let-7b led to increased pro-inflammatory cytokines release in nonpathogenic *E. coli* K-12-infected cells suggesting that through the inhibition of let-7b, AIEC induce the overexpression of TLR4 that triggers an excessive inflammatory response to AIEC and to nonpathogenic bacteria.

#### 5.4.2. Through Survival and Replication within Macrophages

Macrophages are key cells of the innate immune system, they can recognize MAMP (LPS, flagella, and peptidoglycan) through pattern recognition receptors (PRR) such as TLRs. The interaction between TLR and MAMP activates the NF-κB, MAPK, and IRF pathways leading to the expression of genes coding for pro-inflammatory cytokines such as TNF-α or IL-6. Macrophages also limit bacterial dissemination through phagocytosis and through presentation of antigens to activate adaptive immune cells [97]. Macrophages are found in greater numbers in inflamed tissue in CD patients [98], and as mentioned above, AIEC bacteria are able to survive and to replicate within J774 macrophages, in phagolysosomes, without inducing cell death [99] through the induction of the stress gene *htrA*, the oxidoreductase *dsbA* gene, and *gipA* gene in the phagocytic vacuole. Interestingly, the induction of these genes was not observed in a nonpathogenic *E. coli* bacteria, demonstrating that the AIEC LF82 genetic background is crucial for induction of these genes transcription during phagocytosis [100,101,102]. Moreover, AIEC subjected to severe stresses within macrophages show heterogenous stress responses. Indeed, AIEC LF82 either survived in phagolysosomes or underwent phenotypic switches, these responses requiring the induction of SOS responses. Phenotypic switches generated a population of nongrowing bacteria of which a portion presented antibiotic tolerance, demonstrating that intracellular stresses can create phenotypic heterogeneity within AIEC LF82 promoting the production of persister [103]. Therefore, these data demonstrate that AIEC bacteria have evolved to survive and replicate within macrophages through the acquisition of specific regulation of essential genes and essential stress responses for their survival under phagolysosomal conditions.

AIEC-infected macrophages release a large amount of TNF-α and do not undergo cell death [99]. In vitro, the treatment of macrophages with TNF-α promoted the intracellular replication of AIEC, whereas incubation of macrophages with an anti-TNF-α antibody impaired the intracellular replication [104]. Hence, depending on the quantity of intracellular AIEC, infected macrophages secrete different amounts of TNF-α, which in turn increase the intracellular replication of AIEC, demonstrating that targeting TNF-α could be an effective strategy to control the TNF-α-dependent AIEC replication within macrophages (Figure 1).

Briefly, AIEC survive and replicate within macrophages because they develop mechanisms to resist to phagolysosomal conditions and promote their own replication through the stimulation of TNF-α secretion by macrophages.

#### 5.4.3. Through the Release of Exosomes

Exosomes are small membrane vesicles with size ranging from 50 to 100 nm which are freed upon fusion of multivesicular bodies, containing intraluminal vesicles, with the plasma membrane. They are released from many cell types including IECs and immune cells and function in cell-to-cell communication by transferring genetic material such as noncoding RNA and proteins from a donor cell to a recipient cell [105]. Exosomes play a role in immune regulation as they participate in antigen presentation, T-cell activation, and immune suppression, and they have already been involved in infectious diseases [106,107]. Indeed, macrophages infected with intracellular pathogens such as *Mycobacterium tuberculosis*, *Salmonella typhimurium,* or *Toxoplasma gondii* release exosomes containing MAMP which stimulate a pro-inflammatory response when exposed to uninfected macrophages in vitro and in vivo [108]. Carrière et al. studied AIEC-infected IECs and macrophages-released exosomes and the impact of these exosomes on naive cells. In vitro, AIEC infection of human IECs (T84 cells) induced the release of a larger number of exosomes compared to uninfected cells and cells infected with a nonpathogenic *E. coli*. These exosomes from AIEC-infected cells were able to induce a pro-inflammatory response in naive T84 cells and naive THP-1 macrophages through the activation of the NF-κB and the MAPK pathways and the production of pro-inflammatory cytokines [109] (Figure 1). Thus, we see that upon AIEC infection, IECs let exosomes loose that can trigger, in a paracrine effect, a pro-inflammatory response in neighboring IECs but also in macrophages. A similar pro-inflammatory response was observed when naive THP-1 macrophages were exposed to exosomes freed from AIEC-LF82-infected THP-1 cells. Moreover, exosomes released from infected cells can affect AIEC LF82 intracellular replication. Naive T84 or THP-1 cells stimulated with exosomes released from uninfected IECs or macrophages presented decreased AIEC LF82 intracellular replication, whereas the intracellular replication of AIEC LF82 was increased in naive cells stimulated with exosomes released from infected T84 or THP-1 cells. Exosomes released in the ileal lumen of AIEC LF82-infected CEABAC10 mice were able to trigger a pro-inflammatory response in naive recipient macrophages. These results suggest that AIEC are able to hijack the exosomes release to promote their own replication and to promote a pro-inflammatory response in neighboring cells.

AIEC bacteria can, in this manner, induce a pro-inflammatory response in host cells through the activation of different receptors and pathways leading to the secretion of pro-inflammatory cytokines, such as TNF-α, promoting their intracellular replication within macrophages.

### 5.5. Control of AIEC Intracellular Replication

#### 5.5.1. Role of Autophagy in the Control of AIEC Replication

Autophagy is a process, conserved in eukaryotes, leading to the degradation of cytoplasmic materials inside lysosome. Autophagy can be induced under stress conditions like starvation to overcome nutrient deficiency by recycling some components of the cell but it can also be induced to degrade aggregated proteins, damaged mitochondria, or invading pathogens in nonstarved cells. Accordingly, by defending the host against intracellular pathogens, autophagy contributes to the maintenance of the intestinal homeostasis [110]. Autophagy has been involved in IBD etiology following the discovery of susceptibility polymorphisms in autophagy-associated genes. In mouse embryonic fibroblasts (MEFs) and epithelial cell lines, AIEC infection induced autophagy leading to the capture of AIEC by the autophagy machinery and consequently, to the limitation of AIEC intracellular replication [111]. However, in cells knockdown for ATG16L1, protein required for the formation of autophagosome, or IRGM, necessary for the initiation of autophagy, autophagy was impaired and AIEC LF82 replicated intracellularly. The same results were observed in cells expressing the ATG16L1 CD-associated allele. Moreover, the replication and survival of other *E. coli* strains were not impacted in autophagy-deficient cells (*Atg5*^−/−^). These results demonstrate that autophagy specifically limits AIEC survival and intracellular replication in epithelial cells and that CD-associated variant impairs this regulation leading to the persistence of AIEC (Figure 1). The activation of the EIF2AK4-EIF2A-ATF4 autophagy pathway by AIEC infection led to the binding of ATF4 to the promoters of several autophagy genes and to the induction of their expression leading to bacterial clearance and controlled inflammatory response [112]. *In vivo,* following AIEC infection, *eif2ak4*^−/−^ mice did not activate autophagy in IECs and presented increased AIEC intestinal colonization and inflammation compared to WT mice. This pathway was observed as downregulated in ileal biopsies of inflamed patients compared to noninflamed CD patients demonstrating that activation of this pathway and thus, autophagy contributes to controlling the AIEC intracellular replication, while defects in the activation of this pathway could contribute to the active disease.

NOD2, a gene associated with CD, is a cytoplasmic PRR recognizing muramyl dipeptide of bacteria which has been linked to autophagy since it recruits ATG16L1 to the plasma membrane at the site of bacterial entry, leading to the activation of autophagy [113,114]. CD-associated *NOD2* variants impair this response. In NOD2 overexpressing Caco-2 cells, AIEC infection led to a stronger increase in ATG16L1 protein expression associated with a decrease in AIEC survival and of pro-inflammatory cytokines release compared to WT Caco-2 cells [115]. These results highlight the important role of NOD2 in the activation of autophagy and consequently in the promotion of AIEC clearance. Another protein important in the regulation of NOD2 and autophagy is the intermediate filament protein vimentin expressed on the surface of mesenchymal cells. This protein is an important regulator of NOD2 as vimentin interacts with NOD2 through the leucin-rich repeat (LRR) domain of NOD2 allowing its recruitment to the plasma membrane. This interaction is disrupted with CD-associated variants of NOD2 which leads to a cytosolic mislocalization of NOD2 associated with the inhibition of NOD2-dependent autophagy, favoring AIEC survival in vitro [116,117].

Monocyte-derived macrophages (MDM) from CD patients present higher levels of internalized AIEC and are unable to restrict AIEC intracellular replication, leading to disordered inflammatory response compared to MDM from ulcerative colitis patients or healthy controls [118,119]. In THP-1 macrophages, AIEC LF82 infection induced the recruitment of the autophagy machinery at the site of entry of the bacteria and functional autophagy limited AIEC replication [120]. However, impaired expression of CD-associated genes *ATG16L1* or *IRGM* in THP-1 macrophages and *NOD2* in NOD2^−/−^ murine peritoneal macrophages led to an increased AIEC intracellular replication and to the secretion of IL-6 and TNF-α upon AIEC LF82 infection [120]. A more recent study demonstrated that macrophage defect to control intracellular AIEC in CD was linked to autophagy-related polymorphisms in *IRGM* and *ULK-1* genes [121]. Therefore, in CD patients, AIEC might take advantage of autophagy deficiency to persist and replicate in macrophages leading to increased pro-inflammatory response.

#### 5.5.2. Role of micro-RNAs in the Control of AIEC Replication

It is recognized that miRNAs can play a role in CD susceptibility, and subsets of dysregulated miRNAs have been identified in non-inflamed colonic mucosa of IBD patients suggesting their potential involvement in CD, in the onset or relapse of inflammation [94,122]. Brest et al. demonstrated that the miR-196 was overexpressed in inflammatory intestinal epithelium of CD patients and correlated with decreased expression of IRGM*. In vitro*, this miRNA downregulated IRGM but not the risk-associated variant, demonstrating the allele specificity of miR-196 [123]. Lu et al. demonstrated that, in vitro, the miRNAs MIR106B and MIR93 downregulated ATG16L1 leading to the inhibition of autophagy and to an impairment in the removal of AIEC in epithelial cells, whereas miRNAs inhibitors rescued the expression of ATG16L1 and autophagy [124]. Another study demonstrated that AIEC infection upregulated levels of miR-30c and miR-130a in T84 cell line through the activation of the NF-κB pathway. miR-30c and miR-130a, respectively, targeted ATG5 and ATG16L1 that led to the inhibition of autophagy and increased AIEC intracellular replication [125]. In CEABAC10 mice, AIEC infection induced the same effects, and the inhibition of murine miR-30c and miR-130a restored the expression of ATG5 and ATG16L1 leading to efficient autophagy and reduced AIEC intracellular replication and pro-inflammatory response. This study suggests that AIEC could reduce autophagy in host cells through the manipulation of miRNAs pathway to favor their intracellular survival (Figure 1). Moreover, in both studies, an inverse correlation between the levels of MIR160B and ATG16L1 [124] and between miR-30c, miR-130a, and ATG5, ATG16L1 [125] in the mucosa of CD patients has been observed strengthening the link between miRNAs and autophagy. Furthermore, a recent study analyzed the content of exosomes to better understand the increase in AIEC replication in cells receiving exosomes released from AIEC-infected cells. This work demonstrated that the miRNAs miR-30c and miR-130a, inhibiting autophagy, were increased in the exosomes released from AIEC LF82-infected cells compared to exosomes released from uninfected cells. These miRNAs were transferred *via* exosomes to recipient cells and targeted ATG5 and ATG16L1 inhibiting autophagy and promoting AIEC intracellular replication [126]. Hence, AIEC could also inhibit autophagy in neighboring cells through exosomes-dependent transfer of miRNAs from infected to recipient cells.

#### 5.5.3. Role of SUMOylation in the Control of AIEC Replication

AIEC can also modulate autophagy by impairing SUMOylation, a post-translational modification in which ubiquitin-like polypeptides of 10 kDa, SUMOs, are linked to target proteins [127]. SUMOylation of protein seems to be necessary to control intracellular survival of pathogens such as *Shigella flexneri* and *Listeria monocytogenes* [128,129,130]. A recent study investigated whether AIEC infection impacted SUMOylation of IECs. In vitro, inhibition of SUMOylation inhibited autophagy leading to an increased AIEC intracellular replication and AIEC-induced inflammation demonstrating the important role of SUMOylation to control intracellular AIEC *via* autophagy. AIEC infection induced a decrease in the levels of SUMO-conjugated proteins in T84 cells and in IECs of CEABAC10 mice through the upregulation, upon infection, of miR-18 that targeted *PIAS* mRNA encoding a protein involved in SUMOylation [131]. The inhibition of this miRNA restored the expression of *PIAS* and limited the AIEC intracellular replication demonstrating that AIEC bacteria are able to manipulate host SUMOylation to inhibit autophagy in order to favor their intracellular replication.

In conclusion, autophagy is an important player in the control of AIEC colonization by preventing intracellular replication. Genetic polymorphisms as well as AIEC themselves, through the upregulation of miRNAs or the manipulation of host SUMOylation, can impact its efficacy and promote AIEC persistence in the gut of CD patients.

## 6. AIEC in the Lumen

As mentioned above, the presence of AIEC in healthy controls implies that a specific environment is necessary to favor AIEC implantation within ileal mucosa of CD patients. In the lumen, AIEC are not only in contact with the gut microbiota but also with diverse nutrients and molecules, creating a specific intestinal environment that can impact the intestinal microbiota and favors AIEC virulence and colonization. 

### 6.1. Intestinal Environment Favoring AIEC Virulence and Colonization

#### 6.1.1. Regulation of Virulence Genes by Luminal Molecules

The expression of the virulence gene *lpf* coding for LPF and the expression of the *gipA* gene are promoted by the presence of bile salts in the growth medium facilitating the interaction of the bacteria with PP [66,102]. Moreover, it has been demonstrated that the presence of bile salts induced ethanolamine metabolism in AIEC LF82 strain and that the bacteria was able to use ethanolamine as a nitrogen source conferring a competitive advantage for AIEC strains in the intestinal environment [132,133] (Figure 1). Duboc et al. demonstrated a fecal dysmetabolism of bile acids in IBD patients linked to dysbiosis. IBD-associated microbiota presented impaired activities of deconjugation, transformation, and desulfation leading to increased levels of primary conjugated bile acids and decreased levels of secondary bile acids, the latter presenting anti-inflammatory effects in vitro [134]. Hence, bile salts seem to constitute environmental signals that might be used by AIEC to promote their colonization and seem to represent a selective advantage increasing AIEC fitness in the gut. Based on the work of Duboc et al.*,* we can hypothesize that these signals might be mediated by primary bile acids following dysmetabolism in CD patients.

Bacteria of the intestinal microbiota can produce short-chain fatty acids (SCFA) through the degradation of undigested polysaccharides and resistant starches. The major SCFA produced are butyrate, acetate, and propionate and are known for their anti-inflammatory effects [135,136]. Besides, propionic acid (PA) is commonly used in agriculture because of its antimicrobial properties [137,138]. Yet, a recent study demonstrated that exposure of AIEC to PA in vitro and in vivo (through PA supplementation in drinking water of mice) exacerbated the AIEC phenotype (increased ability to adhere to, to invade IECs, and to form biofilms) and promoted AIEC colonization and long-term persistence in vivo (Figure 1). RNA-seq analysis revealed that PA induced a specific transcriptomic program, with induction of genes involved in biofilm formation, stress responses, metabolism, membrane integrity, and transport of alternative carbon sources. Interestingly, the virulence phenotype induced by PA was reversed after the removal of PA. These results not only highlight the potential risks of the use of PA as antimicrobial, especially in the context of AIEC colonization, but also show that AIEC evolved to take advantage of the molecules found in GI tract to outcompete with microorganisms of the microbiota [139].

#### 6.1.2. Influence of the Diet on the Ability of AIEC to Colonize in Intestinal Mucosa

The gut environment can also be impacted by the diet, especially in people living in developed countries having a Western lifestyle. Western diet, enriched in total fat, animal proteins, n-6 polyunsaturated fatty acids, and refined sugars, have been associated with a high risk to develop CD [12,13,14]. The object of several studies was to better catch the impacts of a HF/HS diet on the intestinal microenvironment composition and on AIEC infection. In CEABAC10 mice, a HF/HS diet (used as a Western diet) induced dysbiosis with a distinctive enrichment in *E. coli* population. Moreover, this diet increased intestinal permeability, decreased mucus layer thickness, and decreased SCFA concentration and SCFA receptor GPR43 expression. Hence, the HF/HS diet induced a low-grade inflammation and facilitated colonization of the gut mucosa by AIEC in a microbiota-dependent manner [81,140] (Figure 2). 

CD patients frequently present deficiencies in methyl-donor molecules such as folate (vitamin B9) and vitamin B12. As we identified the AIEC receptor gene *CEACAM6* as regulated in a DNA methylation-dependent manner, we suggested that methyl-deficient diet could favor its expression and AIEC intestinal colonization. *CEACAM6* promoter was hypomethylated in CEABAC10 transgenic mice colonic mucosa fed a low methyl diet leading to an abnormal gut expression of the CEACAM6 receptor favoring AIEC colonization [60] (Figure 2). These observations show that the diet composition can modulate key gene expression in the host, favoring the encroachment of AIEC bacteria to intestinal mucosa.

#### 6.1.3. Influence of the Nutrients and Carbon Sources on the Fitness of AIEC

Kitamoto and colleagues recently reported that inflammation-induced blooms of *E. coli* pathobionts was significantly blunted when amino acids, particularly L-serine, were removed from the diet. The authors reported that the metabolism of AIEC shifted toward the catabolism of L-serine in the inflamed gut so as to maximize AIEC growth potential and to outcompete with commensal bacteria, which was not observed in noninflammatory condition [141] (Figure 2). Interestingly, Elhenawy et al. assessed the adaptive evolution of AIEC in a murine model of chronic colonization across multiple hosts and transmission events. In this model, an ancestor AIEC strain was transited in the gut of different mice to study its evolution under gastrointestinal conditions, and it led to the selection of isolates able to use acetate, a SCFA, as a carbon source. This isolate presented a faster generation time and outcompeted the ancestor strain in co-infected mice demonstrating that the use of acetate increases AIEC fitness in vivo [49] (Figure 1). Moreover, improved acetate utilization is more common among *E. coli* isolates from CD patients compared to healthy controls suggesting that within the host, AIEC evolved differently than commensal strains and acquired metabolic advantage.

As regards all these observations, the effect of diet on intestinal homeostasis and on AIEC virulence shows that dietary intervention, modifying the availability of luminal nutrients, could be used, along with pharmacological treatments, to limit AIEC overgrowth and implantation in intestinal mucosa of CD patients. At this time, exacerbation of AIEC phenotype has only been demonstrated with PA, but similar effects might be observed with other dietary molecules, notably polluting molecules such as titanium dioxide, TiO_2_, or dietary emulsifiers. To study the effect of these compounds on AIEC virulence would be noteworthy, as they appear to be largely present in processed food and knowing that some detrimental effects of emulsifiers on the intestinal microbiota have already been described [142,143].

### 6.2. Impact of AIEC Infection on Microbiota Composition

AIEC can take advantage of the presence of a dysbiosis to colonize. For example, the use of antibiotics, leading to dysbiosis, in a mouse model mimicking CD susceptibility is needed to obtain AIEC colonization in vivo [29]. Moreover, *in vivo,* prior AIEC colonization worsened the inflammatory response and disease outcome following *Salmonella* Typhimurium infection. Interestingly, this was related with the expansion of the AIEC population and a delay in the recovery, which indicates that people carrying AIEC might be at a greater risk to develop CD in case of an acute gastroenteritis [144]. 

AIEC can also detrimentally impact the intestinal microbiota. As an example, TLR5-deficient (T5KO) mice developed spontaneous colitis in part due to the inability to manage proteobacteria [145,146]. In this genetically predisposed mouse model, infection of germ-free mice with the AIEC reference strain LF82 drove chronic gut inflammation associated with alterations in gut microbiota composition including a loss in species diversity. Moreover, increased levels of flagellin and LPS in the feces have been noticed compared to WT and uninfected T5KO mice, flagellin and LPS having a great potential to activate innate immune response [30]. These microbiota modifications seemed independent of intestinal inflammation suggesting that AIEC might instigate colitis in T5KO mice following alterations of microbiota composition that increase its pro-inflammatory potential. To improve the understanding of how microbiota alterations might trigger colitis, Chassaing et al. used WT and T5KO mice harboring the “Altered Schaedler Flora” (ASF) microbiota, a community of eight microorganisms devoid of pathogenic/pathobiont bacteria that coexist in the gut and favor the development of normal intestinal immune system [147]. Unlike WT ASF mice, AIEC-infected T5KO ASF mice presented alterations in microbiota composition associated with elevated levels of LPS and flagellin, and these mice developed only a modest low-grade inflammation compared to the previous study [148]. This result confirms the fact that AIEC can alter microbiota composition but suggests that a complex dysbiotic microbiota is necessary for the development of robust inflammation.

Another mouse model exhibiting autophagy defects (Tg/eif2ak4^-/-^) presented an alteration of microbiota composition. In this model, AIEC infection resulted in modifications of microbiota composition and development of gut inflammation [149]. However, alterations of the microbiota composition preceded the development of the gut inflammation demonstrating that in genetically predisposed hosts, AIEC infection and colonization may alter microbiota composition favoring the development of intestinal inflammation.

Thence, the intestinal environment, modulated by luminal molecules and by the diet, is an important actor during AIEC colonization. It can promote the implantation of AIEC directly by driving the expression of different virulence factors or indirectly by impacting the composition of the intestinal microbiota, not to mention that AIEC themselves can detrimentally alter the intestinal microbiota. All these factors promote the persistence of AIEC in the intestinal lumen.

## 7. Targeting AIEC

The last part of this review will focus on presenting and discussing recent therapeutic strategies, aiming at preventing AIEC implantation/persistence, which have shown beneficial effects in preclinical models of CD or which are currently tested in clinical trials.

### 7.1. Inhibition of AIEC–IECs Interaction

#### 7.1.1. Probiotics

Probiotics are living microorganisms having a beneficial impact on the health when administered in adequate amounts, whereas prebiotics are dietary compounds inducing the growth or activity of beneficial microorganisms. AIEC survival and growth were decreased upon coculture with *Lactobacillus rhamnosus GG, L. reuteri* 1063, and the prebiotic inulin, whereas the prebiotic arabinoxylans lowered AIEC adhesion to mucin in vitro [150]. In the simulated mucosal environment (M-SHIME), two compartments exist: a luminal compartment and a mucosal compartment due to the addition of mucin-covered microcosms in the suspension, and in this way, generating a luminal and mucosal microbiota. *L. reuteri* 1063, arabinoxylans, and inulin decreased the colonization of AIEC in the mucus, whereas AIEC numbers in the luminal environment were unchanged. These results were associated with some changes in the microbiota composition of the mucus. Thence, the mucosal microbiota might be modulated by pro- and prebiotic in vitro to decrease AIEC colonization in the mucus and thus possibly limiting AIEC interaction with host cells. However, further studies should be performed to determine if similar results are obtained in in vivo models and if administration of such prebiotic or probiotic strains in CD patients might be a suitable therapeutic approach.

Probiotics able to inhibit the interaction between AIEC and host cells might represent a good therapeutic option for AIEC-positive CD patients. Preincubation of Intestine-407 cells with the probiotic strain *E. coli* Nissle 1917 prior to infection with AIEC strains or coincubation of *E. coli* Nissle 1917 with AIEC strains strongly inhibited the ability of AIEC to adhere to and to invade Intestine-407 cells, highlighting that *E. coli* Nissle 1917 could be efficient in patients with CD to prevent or limit AIEC colonization [151]. Sivignon et al. studied the probiotic effect of the yeast *Saccharomyces cerevisiae* CNCM I-3856 on AIEC ability to interact with IECs. Yeasts present a high content of mannose residues on their cell wall that can be recognized by type 1 pili of bacteria. *S. cerevisiae* CNCM I-3856 strongly inhibited adhesion of AIEC LF82 to IECs in culture and to isolated enterocytes from CD patients, through the type-1 pili-mediated agglutination of AIEC bacteria to the yeast [152]. In CEABAC10 mice infected with AIEC LF82, *S. cerevisiae* CNCM I-3856 and yeast derivatives decreased AIEC gut colonization and prevented the increase of intestinal permeability and the release of pro-inflammatory cytokines. These results demonstrate that *S. cerevisiae* CNCM I-3856 presents beneficial effects which are not dependent of its viability but which are possibly driven by the inhibition of the FimH/CEACAM6 interaction (Figure 3). Hence, *S. cerevisiae* could be a good therapeutic option for CD patients already colonized or susceptible to be colonized by AIEC.

#### 7.1.2. Inhibition of AIEC Adhesion Using Chemical Compounds

The FimH-CEACAM6 interaction, in which the FimH adhesin interacts with the mannose residues of the CEACAM6 receptor expressed at the surface of IECs, is one of the most studied since it favors AIEC gut implantation. Therefore, researches focused on the development of antiadhesive molecules to saturate the carbohydrate recognition domain of FimH and thus inhibiting this interaction to limit AIEC colonization. For example, thiazolylaminomannosides and *n*-heptyl α-D-mannose (HM)-based glycopolymers presented high affinity for FimH and were 10^2^ and 10^4^–10^6^ times more potent than HM and D-mannose (well-known antagonists of FimH), respectively, to inhibit AIEC LF82 adhesion to IECs *in vitro. Ex vivo,* these compounds also prevented AIEC LF82 adhesion to colonic tissue of CEABAC10 at very low concentrations demonstrating their strong antiadhesive potential [153,154,155]. Sivignon et al. additionally demonstrated that heptylmannoside derivatives strongly impaired the ability of adhesion of AIEC LF82 to T84 cells and reduced the amount of AIEC in feces and associated to the mucosa in AIEC LF82-infected CEABAC10 mice. Moreover, treatment with these compounds decreased the severity of colitis and intestinal inflammation induced by AIEC LF82 in these mice. Heptylmannoside derivatives are shown as presenting strong antiadhesive effects in vitro and protect in vivo against colitis implying that they could be interesting compounds to treat AIEC-colonized CD patients [156]. Similar results were obtained with other analogues of HM and heptylmannoside grafted on cellulose nanofibers [157,158]. Targeting FimH with antiadhesive molecules is consequently a promising method to limit AIEC adhesion to IECs (Figure 3). This strategy is currently under evaluation in a clinical trial using the FimH blocker EB8018/TAK-018 molecule to evaluate its effect on postoperative endoscopic recurrence in 96 postoperative participants with CD (phase 2a, NCT03943446). 

#### 7.1.3. Other Strategies

IECs express several types of endo- and exoproteases to protect the host against bacterial colonization. Particularly, meprin induced the proteolytic cleavage of type 1 pili of AIEC LF82 bacteria that impaired their ability to bind mannosylated residue, like those present on host receptors, explaining the reduced abilities of AIEC LF82 strain pretreated with meprin to adhere to and to invade T84 cells [159]. Moreover, levels of meprin are decreased in patients with ileal CD. The increase of the level of meprin in CD patients through the administration of purified meprin or through meprin-producing probiotic strains could then be an interesting strategy to disrupt the FimH-CEACAM6 interaction to limit AIEC adhesion to IECs and gut colonization (Figure 3). 

Moreover, the OmpA-Gp96 interaction promotes AIEC invasion through the delivery of proteins into the host cells [70,71]. A good therapeutic option to inhibit AIEC colonization could be in this way to block this interaction. An antagonist of Gp96 exists, the synthetic peptide Gp96-II, and protected mice against intestinal inflammation in vitro and in vivo [160]. It could be interesting to test this antagonist in a context of AIEC infection to determine if it leads to the disruption of OmpA-Gp96 interaction and to the inhibition of AIEC colonization in vitro and in vivo while seeking for new Gp96 antagonists at the same time (Figure 3).

In intestinal mucosa of CD patients, increase in relative abundance of well-known mucus degraders, particularly *Ruminococcus gnavus,* which secretes glycosidases, is frequently observed [161]. In CEABAC10 mice, Western diet induced a dysbiosis, comparable with what is observed in CD patients, with an increase in the mucin-degrading bacterium *Ruminococcus*, allowing AIEC bacteria to better colonize the gut mucosa [81]. Therefore, mucins degradation in the gut may rely on the cooperative action of several microbial species characterized as “mucin-degraders.” Interestingly, AIEC secrete a serine mucin-protease, Vat-AIEC, which is more frequently found in AIEC than in non-AIEC strains and promotes AIEC encroachment in the gut, leading in this manner to dysbiosis and chronic inflammation [46]. It turns out that developing new antivirulence strategies, such as mucinase inhibitors, could also represent an effective therapeutic option to better manage CD patients colonized by AIEC (Figure 3).

### 7.2. Elimination of AIEC

#### 7.2.1. Antibiotics

Antibiotics appear like the first logical approach to manipulate the microbiota of CD patients and eliminate AIEC, and several antibiotics have been tested to treat CD such as ciprofloxacin, aminoglycosides, or rifaximin [162,163] (Figure 4). A study revealed in an interesting way that rifaximin reduced the adhesion and invasion abilities of AIEC strains to colonic T84 epithelial cells and limited the pro-inflammatory cytokine IL-8 secretion, through regulation of virulence gene expression and motility, independently of its antimicrobial effect [164]. Based on these preclinical data, a double-blind randomized clinical trial (THEOREM, phase 2, NCT02620007) aiming at evaluating the effect of a 12-week treatment with ciprofloxacin and rifaximin on endoscopic remission in AIEC-colonized patients with ileal CD is currently ongoing.

#### 7.2.2. Use of Colicin

Colicins are the best-characterized family of bacteriocins, namely, highly selective species-specific antibiotics. Colicins are produced by Gram-negative bacteria such as *Escherichia coli* strains carrying a colicinogenic plasmid. These colicins are able to kill other bacteria through a nuclease activity, pore-forming activity, or through the inhibition of cell wall synthesis [165,166,167]. Purified colicin E1 and E9 as well as a commensal *E. coli* strain naturally producing E1 colicin presented killing activity against AIEC LF82 biofilms. These E1 and E9 colicins also induced the killing of AIEC bacteria adhering to IECs and of intramacrophagic bacteria in vitro without inducing cell toxicity. The intramacrophagic effect was due to the fact that colicins have access to AIEC-containing compartments within infected macrophages through actin-mediated endocytosis [168]. Hence, colicins might represent a good therapeutic approach to selectively target AIEC in CD. They could be delivered as a purified protein or through a colicin-producing probiotic, an *E. coli* Nissle 1917 strain producing colicin already existing [169] (Figure 4).

#### 7.2.3. Phagotherapy

Bacteriophages are viruses able to infect bacteria and are found in diverse environments including the human gastrointestinal tract [170,171]. Bacteriophages are highly specific; they only target a limited number of bacterial strains within given species which limits their impact on the composition of the intestinal microbiota and so far, no evidence that phage therapy causes adverse effects have been reported [172,173]. From three bacteriophages isolated from waste water and able to infect AIEC in vitro and ex vivo, Galtier et al. investigated their potential to decrease AIEC intestinal colonization in vivo (Figure 4). A single-dose bacteriophage cocktail reduced the gut colonization of AIEC LF82 in CEABAC10 mice and in DSS-treated WT mice, bacteriophage treatment reduced the ileal and colonic colonization of AIEC LF82 and decreased the symptoms of DSS-induced colitis over a 2-week period. Lastly, bacteriophages were able to replicate ex vivo on ileal biopsies from CD patients infected with AIEC LF82 demonstrating their potential efficacy on intestinal mucosa [174]. The results from an ongoing clinical trial targeting AIEC bacteria in CD patients are expected in June 2021. This phase 2, double-blind, placebo-controlled clinical trial aims at studying the effect of an AIEC-specific bacteriophage cocktail (EcoActive) on disease activity, inflammatory markers (CRP and fecal calprotectin), and AIEC load in a cohort of 30 CD patients (NCT03808103).

#### 7.2.4. Bacterial Predation/Competition

##### Bacterial Predation

*Bdellovibrio bacteriovorus* bacterium is a predator of Gram-negative bacteria playing the role of “ecological balancer” in several habitats, and it highly colonizes the intestinal mucosa, especially duodenum and ileum, of healthy human subjects compared to IBD patients [175,176]. In fact, *B. bacteriovorus* migrates and encounters prey bacteria through its long flagellum, then invades the periplasmic space of the prey and replicates through a complex cycle leading to the death of the prey and the release of its progeny [177,178]. *B. bacteriovorus* demonstrated predatory activity against AIEC LF82, in planktonic culture and on preformed and developing biofilms, and decreased AIEC LF82 adhesion to and invasion of Caco-2 cells (Figure 4). In vivo, larvae of *Galleria mellonella* presented an increased survival rate upon AIEC LF82 infection pre-exposed to *B. bacteriovorus* demonstrating that *B. bacteriovorus* limits pathogenicity of AIEC or the host susceptibility to be colonized by AIEC [179]. Therefore, *B. bacteriovorus,* bacterium present in healthy subjects, could represent an interesting therapeutic strategy to limit AIEC overgrowth in CD patients by exploiting a natural process. However, it remains necessary to identify the consequences of the presence of *B. bacteriovorus* on global microbiota composition, especially on Gram-negative bacteria. Indeed, this bacterium could also broadly target Gram-negative bacteria, which may actually open an ecological niche further for AIEC bacteria colonization.

##### Type VI Secretion System

The type VI secretion system (T6SS) is a nanoweapon used by Gram-negative bacteria to inject toxic effectors directly into eukaryotic cells or bacteria. T6SS resembles a bacteriophage tail-like structure that is anchored to the bacterial cell envelope through a membrane-associated complex. Effectors are delivered in a one-step contact-dependent manner upon contraction of a sheath propelling a syringe puncturing the target cell [180,181]. Some enteric pathogens use T6SS to kill commensal bacteria and colonize the gut, such as *Salmonella* Typhimurium that killed *Klebsiella oxytoca* favoring its implantation in the gut of mice [182]. On another note, T6SS can also be used by commensal bacteria, notably, *Bacteroides fragilis* used T6SS to antagonize commensal Bacteroidales species and to kill enterotoxigenic *B. fragilis* strains in vivo in mice [183,184]. The pathogen *Shigella sonnei* specifically outcompeted *Escherichia coli in vitro* and in vivo in mice in a T6SS-dependant manner [185], whereas a recent study demonstrated that it is possible to manipulate the type VI secretion system of *Pseudomonas aeruginosa* to shuttle chosen proteins, different of canonical effectors [186]. Hence, it would be interesting to discover if some commensal bacteria are able to target AIEC in T6SS-dependent manner in order to develop a highly specific and safe therapeutic option for CD patients. In addition, from the work of Anderson et al. and Wettstadt and Filloux, a hypothesis could be to manipulate the T6SS of a commensal bacterium to deliver toxins specific of AIEC in order to safely eliminate these pathobiont bacteria (Figure 4). As specified above, the use of this strategy should rely on the fact that T6SS specifically targets AIEC bacteria, and not all the Gram-negative bacteria in an unspecific manner, to avoid the bloom of AIEC due to the opening of an ecological niche favorable to AIEC growth. This suggests that the development of strategies specifically targeting AIEC should be encouraged, as proposed below.

##### Conjugative Bacteria and CRISPR

Conjugation is one of the most efficient horizontal gene transfer mechanisms used by bacteria to spread pathogenicity islands and antibiotic-resistant genes to neighboring bacteria. A conjugative plasmid has been successfully used to transfer the clustered regularly interspaced short palindromic repeat (CRISPR) nuclease machinery (Cas9 enzyme + sgRNA) from *E. coli* to *Salmonella enterica*, leading to *Salmonella* killing [187]. This strategy has also been used to selectively remove antibiotic-resistant genes in *Enterococcus faecalis in vitro* and in vivo [188]. These results clearly show that a conjugative probiotic strain could be used to deliver the CRISPR-Cas9 system to pathobiont bacteria through the use of a RNA guide targeting specific gene of *E. coli* or more specifically, targeting polymorphisms highly prevalent in AIEC strains (FimH pathoadaptive mutations for example). Our hypothesis is that the use of probiotic bacteria, harboring a conjugative plasmid encoding for CRISPR machinery, and specific RNA guide would lead to the specific and controlled killing of AIEC bacteria without altering other members of the microbiota. This would lead to a precise engineering of complex microbiota with the elimination of very specific strains within a multibacterial environment (Figure 4).

### 7.3. Activation of Autophagy

Upon AIEC infection, autophagy is induced and enables to limit intracellular replication of AIEC and therefore their persistence. But, as described earlier, CD-associated variants exist in autophagy-related genes such as *ATG16L1*, *IRGM,* or *NOD2* impairing autophagy and favoring AIEC intracellular replication and persistence in the gut. Lapaquette et al. demonstrated that physiological (starvation) and pharmacological (rapamycin) induction of autophagy prevented AIEC intracellular replication and pro-inflammatory cytokines release in NOD2^-/-^ murine macrophages [120]. Azathioprine, a current IBD drug, activated autophagy in THP-1 macrophages in vitro enhancing the clearance of intracellular AIEC and reduced the pro-inflammatory response. This drug activated also autophagy in peripheral blood mononuclear cells (PBMC) from pediatric patients even in PBMC presenting a CD-associated variant for ATG16L1 [189]. Hence, in CD patients presenting polymorphisms in autophagy-associated genes, activation of autophagy with pharmacological drugs might be an interesting therapeutic solution to limit AIEC persistence and inflammation. In addition, a recent study revealed that resveratrol, a micronutrient present in grapes and wine, boosted autophagy favoring autophagy-dependent clearance of AIEC in vitro in IECs. That would imply that such compound might be also used in CD patients to stimulate autophagy and promote AIEC clearance [190] (Figure 4). On the other hand, it seems important to evaluate the efficacy of pharmacological activation of autophagy in CD patients harboring polymorphisms in autophagy genes before proposing a clinical trial based on this strategy, since these autophagy-activating treatments might not be effective for patients with genetic autophagy defects.

### 7.4. Nutritional Interventions

In this review, we described many environmental factors favoring the encroachment of AIEC bacteria to the intestinal epithelium. Nutritional interventions are of great interest to limit the colonization of AIEC bacteria and to favor the expansion of beneficial bacteria in the microbiota of CD patients. We previously revealed that a methyl-donor-deficient diet (deficient in folate and vitamin B12) led to an overexpression of CEACAM6 favoring AIEC intestinal colonization. Moreover, Kitamoto and colleagues revealed that the metabolism of AIEC shifted toward the catabolism of L-serine in the inflamed gut in order to maximize AIEC growth, and outcompete with commensal bacteria, and that depletion of the serine from the diet limited AIEC overgrowth under inflammatory conditions. In addition, polysaccharides are commonly added to processed foods to improve palatability, and it has been demonstrated that AIEC exposure to maltodextrin, a polysaccharide derived from starch hydrolysis, induced the expression of type I pili enhancing AIEC LF82 biofilm formation and bacterial adhesion to IECs in vitro [191]. 

Based on these observations, one dietary strategy could be to supplement the diet of CD patients harboring AIEC bacteria with methyl donor molecules, to prevent CEACAM6 abnormal expression, and to limit the intake in serine and maltodextrin, to prevent the blooming and gut colonization of AIEC. Prebiotics could also be offered to CD patients to favor the growth and activity of protective bacteria which would limit the implantation of AIEC pathobionts; this therapeutic option being supported by the fact that two prebiotics, arabinoxylans and inulin, can repress AIEC from mucus in mucosa-comprising gut model [150].

### 7.5. Flagellin Vaccination

As already mentioned, increased levels of flagellin in feces are observed upon AIEC infection in genetically predisposed T5KO mice [30]. Flagellin is a known trigger of intestinal inflammatory response through the activation of TLR5 and NLRC4. In this way, stimulating adaptive immunity against flagellin could be a means to reduce AIEC persistence within the gut and to limit the inflammation. Weekly administration of flagellin to WT mice resulted in increased levels of anti-flagellin IgA in the serum and the feces and altered the composition of the gut microbiota associated with increased IgA-coated bacteria, reduced levels of fecal flagellin, and decreased microbiota encroachment [192]. Thus, flagellin administration decreased the pro-inflammatory potential of the microbiota and the presence of flagellated bacteria. Moreover, flagellin administration protected WT mice against a colitis induced by immune dysregulation (injection of IL-10 receptor-neutralizing antibody). Based on these data, it would be interesting to test if the strong anti-flagellin IgA response following administration of flagellin promotes the elimination of AIEC and reduces the inflammatory response in vivo upon AIEC infection (Figure 4). Of note, AIEC bacteria have been shown as highly targeted by IgA in CD-associated spondyloarthritis patients’ microbiota, reinforcing the hypothesis that such a flagellin vaccine-based strategy could benefit CD patients [193]. 

In conclusion, several promising therapeutic approaches exist whose efficacy has been demonstrated in in vitro and in vivo models and some of them are currently under validation in ongoing clinical trials. Further studies need to be conducted to deepen the achieved results and to check the safety of such treatments while new strategies are further proceeded. 

## 8. Future Directions 

Many studies have clearly shown the association between the abnormal prevalence of AIEC bacteria in CD patients and the onset of the disease, mostly in industrialized countries. In contrast, very few studies highlighted the presence of these pathobiont bacteria in “in-development” countries such as in South East Asia and Middle East countries, despite the huge increase in the incidence of IBD during the last 10 years. It seems important to evaluate whether AIEC bacteria are common actors in CD all around the world. 

One important point in the study of AIEC is the phenotypical characterization of these bacteria. To date, this characterization is not well-standardized in all the laboratories around the world. An effort should be done to standardize the experiments aiming at characterizing these strains to obtain robust prevalence test of AIEC in CD population. The phenotypical characterization is a long process and requires time-consuming experiments before concluding whether the isolated bacteria are AIEC or not. Hence, it seems necessary to develop new tool for a simplified and standardized identification of AIEC bacteria. Instead of studying the bacteria itself, AIEC-specific biomarkers expressed by host cells could be identified (which is one of the aims of the MOBIDIC clinical trial). As an example, immune-dominant antigens expressed by AIEC, leading to an important humoral response, could be identified and could help the development of serological assay predictive of the carriage of AIEC in CD patients. In the case a humoral response specific to AIEC bacteria would exist, future researches could focus on the development of vaccinal therapies to prevent AIEC overgrowth.

Finally, preclinical models of CD confirmed the causal or contributing role of AIEC bacteria to the development of CD. However, showing that specifically removing AIEC from intestinal mucosa (as performed in many ongoing clinical trials described above) in CD patients limits the symptoms and relapses of the disease is necessary to validate the role of these bacteria in CD and to propose AIEC targeting as a relevant therapy in CD.

To conclude, efforts to better understand the physiology of these bacteria and to elucidate their exact role in CD need to be pursued. It will not only improve diagnostic approaches but will also offer new therapeutic arsenal to propose personal therapies to CD patients colonized by pathobiont AIEC bacteria.

## Figures and Tables

**Figure 1 ijms-21-03734-f001:**
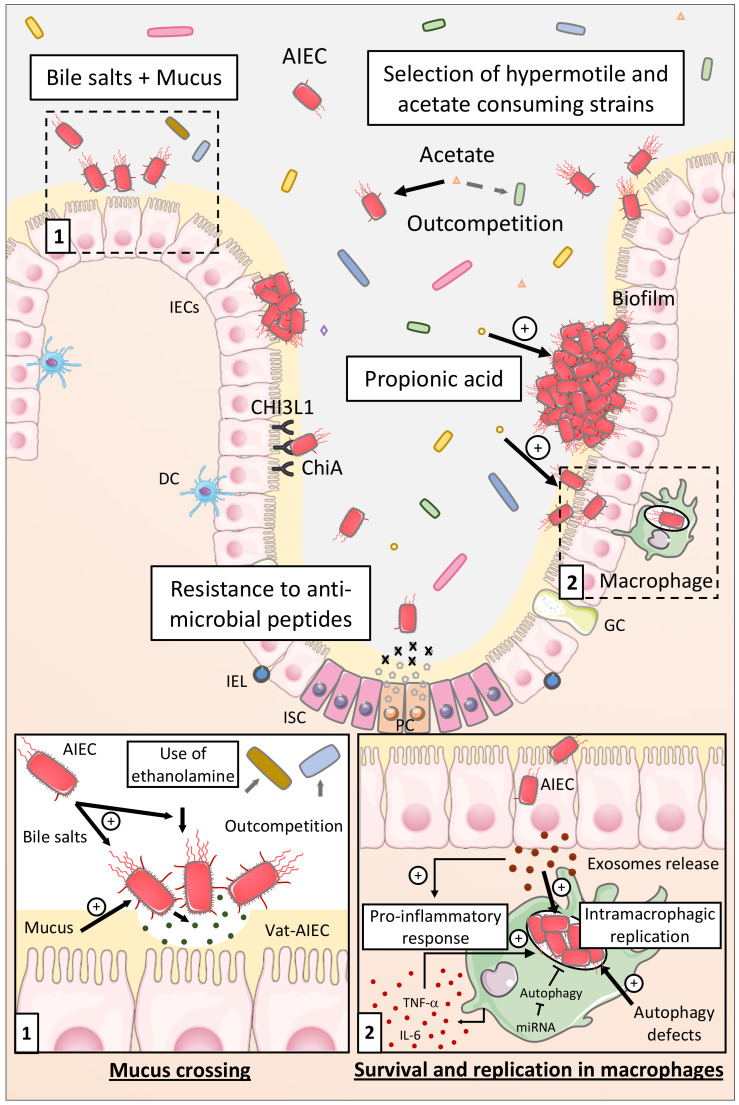
Access to the mucosa and impact of luminal molecules on adherent-invasive *Escherichia coli* (AIEC) virulence and colonization. In the lumen, bile salts and mucus favor the expression of several AIEC virulence genes such as *lfp, fliC*, and vat-AIEC promoting mucus crossing. Bile salts also induce ethanolamine metabolism in AIEC conferring a competitive advantage over commensal bacteria (1). AIEC strains presenting hypermotile phenotype and able to use acetate are selected under gastrointestinal conditions, while the short-chain fatty acids (SCFA), propionic acid, exacerbates AIEC phenotype with increased abilities to form biofilms, to adhere to and invade intestinal epithelial cells (IECs). Conditions promoting AIEC survival and replication within macrophages (2) as well as ChiA-CHI3L1 interaction and AIEC resistance to antimicrobial peptides are also represented. DC, dendritic cell; GC, goblet cell; IECs, intestinal epithelial cells; IEL, intraepithelial lymphocytes; IL-6, interleukin 6; ISC, intestinal stem cell; PC, Paneth cell; TNF-α, tumor necrosis factor alpha.

**Figure 2 ijms-21-03734-f002:**
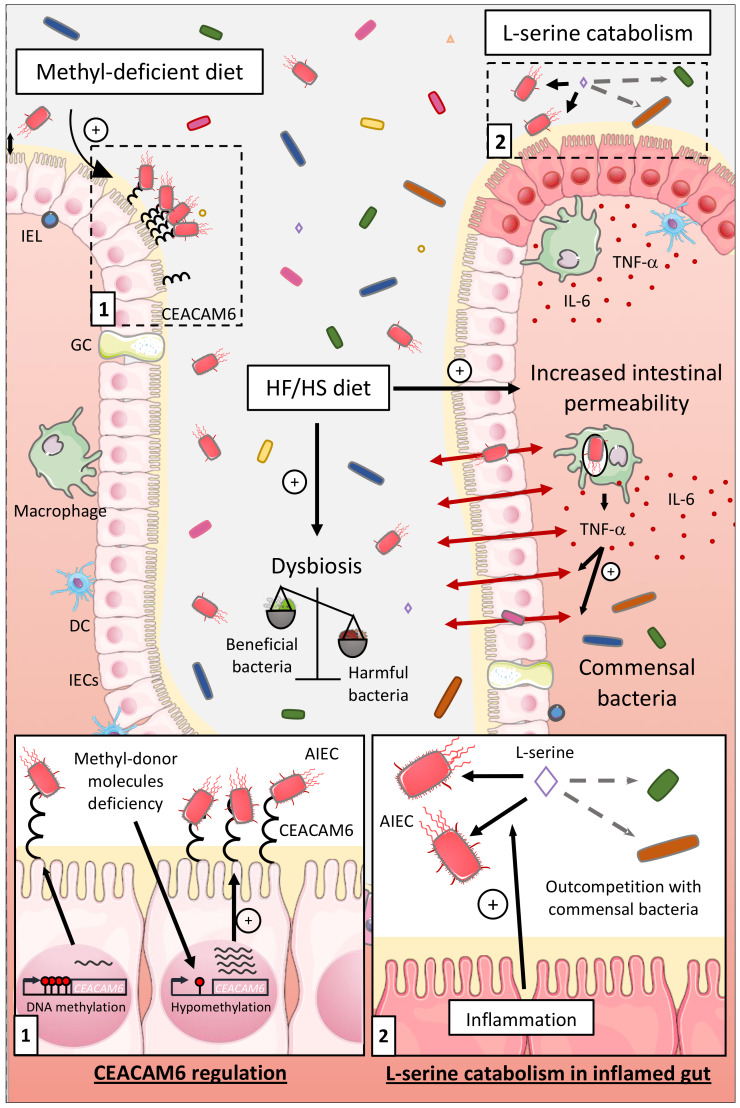
Impact of the diet and intestinal inflammation on AIEC virulence and colonization. Western diet consumption induces dysbiosis, decreases the mucus layer thickness, and increases intestinal permeability favoring bacterial translocation across the epithelial layer. All these effects create a low-grade intestinal inflammation and favor AIEC colonization. In addition, a diet deficient in methyl-donor molecules leads to hypomethylation of the carcinoembryonic antigen-related cell adhesion molecule 6 (CEACAM6) promoter and its abnormal expression promoting AIEC gut implantation (1). The catabolism of L-serine, whose concentration depends on dietary intake, is promoted in inflamed gut increasing AIEC competitive fitness (2). DC, dendritic cell; GC, goblet cell; IECs, intestinal epithelial cells; IEL, intraepithelial lymphocytes; IL-6, interleukin 6; TNF-α, tumor necrosis factor alpha.

**Figure 3 ijms-21-03734-f003:**
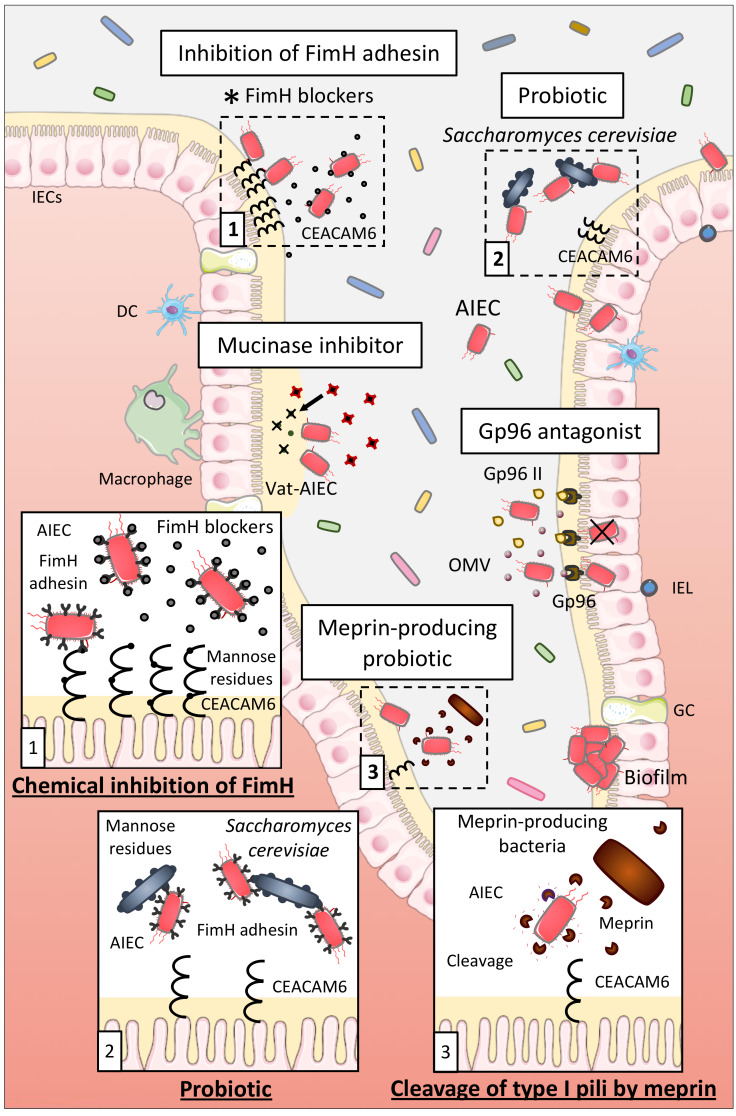
Strategies to prevent AIEC colonization in CD patients. Anti-adhesive molecules inhibiting the FimH adhesin (1) or a probiotic strain such as *Saccharomyces cerevisiae* containing mannose residues in its cell wall (2) can inhibit the FimH–CEACAM6 interaction through competition by binding to bacterial FimH. Meprin-producing probiotic strain could limit AIEC interaction with IECs through the meprin-mediated cleavage of AIEC type I pili (3). Other strategies are also suggested as interesting to prevent AIEC colonization such as the use of Gp96 antagonist to block the OmpA-Gp96 interaction or the use of a mucinase inhibitor to prevent mucus degradation and AIEC encroachment. * Star shows the AIEC targeting approach currently under investigation in a clinical trial. DC, dendritic cell; GC, goblet cell; IECs, intestinal epithelial cells; IEL, intraepithelial lymphocytes.

**Figure 4 ijms-21-03734-f004:**
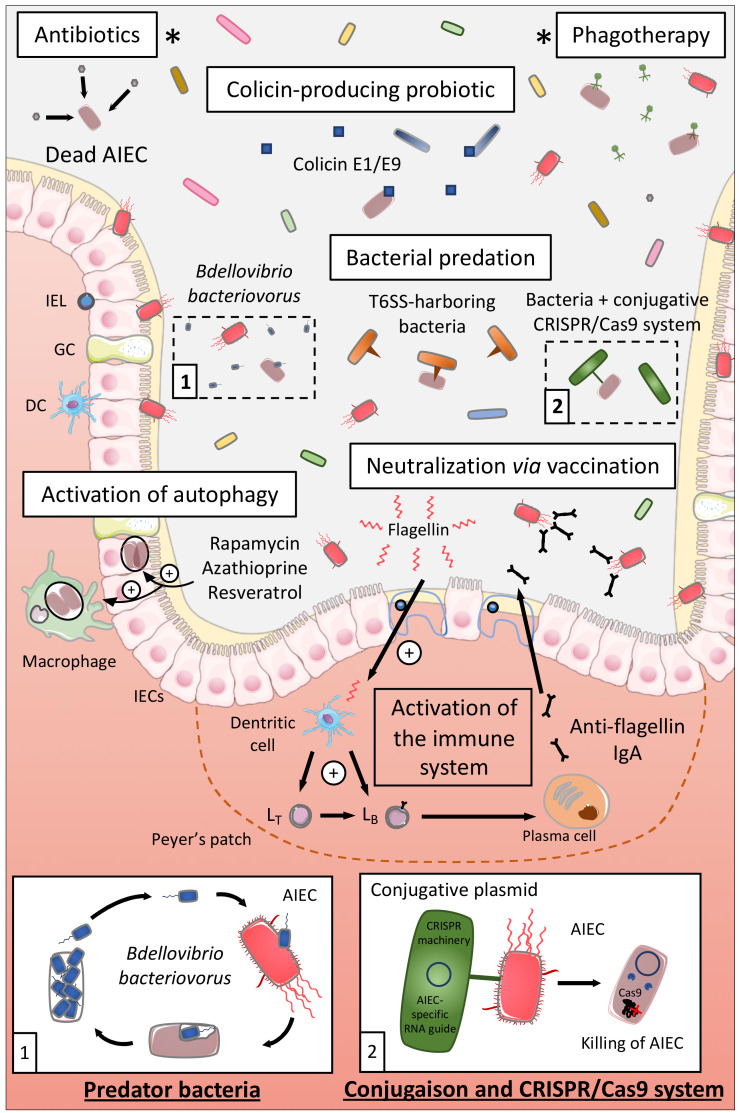
Strategies to eliminate AIEC in Crohn’s disease (CD) patients. AIEC bacteria could be killed by some antibiotics but they could also be eliminated by colicin-producing probiotic strain or bacteriophages specifically targeting AIEC. Some strategies of bacterial predation are also under study where AIEC would be specifically targeted by predator bacteria (1), T6SS-harboring bacteria, or by bacteria harboring the conjugative clustered regularly interspaced short palindromic repeat (CRISPR)/Cas9 machinery with a RNA guide specific of AIEC (2). Activation of autophagy with pharmacological drugs in CD patients are also studied to promote AIEC clearance. Finally, a vaccination approach based on flagellin administration might be a mean to neutralize AIEC bacteria *via* the production of anti-flagellin IgA. * Stars show the AIEC targeting approaches currently under investigation in clinical trials. DC, dendritic cell; GC, goblet cell; IECs, intestinal epithelial cells; IEL, intraepithelial lymphocytes; IgA, immunoglobulin A; L_B_, lymphocyte B; L_T_, lymphocyte T; T6SS, type VI secretion system.

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
