# Peer review of "Adherent-Invasive E. coli: Update on the Lifestyle of a Troublemaker in Crohn’s Disease"

_ijms, 2020, doi:10.3390/ijms21103734_

Round 1

Reviewer 1 Report

This review by Chervy and colleagues is presenting a thorough overview of all the current knowledges regarding adherent and invasive Escherichia coli associated with Crohn’s disease. This article nicely brings together all the past findings regarding AIEC pathogenicity, is illustrated by outstanding figures, and finish by important thoughts regarding the future promises hold by targeting AIEC in Crohn’s disease patients. Moreover, in addition to this list of published virulence factors and mechanisms by which AIEC bacteria can promote chronic intestinal inflammation, the authors also present some interesting open questions and hypothesis throughout the manuscript. I nonetheless have few comments, especially regarding the general organization of the review, as detailed below.

Major comments

- Chapter "2. Virulence factors of the AIEC pathovar" is listing, without giving much details, all known virulence factors which will be subsequently presented in a more detailed manner. Hence, I suggest to delete this chapter that's not bringing anything, expect spoiling the following detailed explanations.

- Chapter 3 - AIEC in the lumen - is treating a lot about regulation of virulence factors, and hence should be moved down, after all these virulence factors are presented in details. This will greatly improve clarity and avoid unnecessary repetitions.

- The figures presented are outstanding, and the authors are thank for this contribution. Well done !

- Regarding the use of Bdellovibrio bacteriovorus, a possibility if that this bacterium will broadly target Gram-negative bacteria, which may actually open the niche further for AIEC bacterium (even if AIEC are also targeted). This should be mentioned, as it illustrate the complexity in reversing a dysbiotic state.

- Same comment for T6SS - which further highlight the promise of CRIPR approach that can be highly specific to AIEC bacteria.

Minor comments

- English need proofreading. For example, "the" have to be deleted when plural (Among the microbiota alterations).

- "Another mouse model exhibiting autophagy defects (Tg/eif2ak4-/-) notified the alteration of microbiota composition." While the use of a mouse model can indicate something, it cannot notice anything.

- Both figures appeared to be duplicated.

Author Response

Dear Editor,

We thank the reviewers for their very pertinent comments, which clearly improved the manuscript. We hope we properly answered to their concerns in the revised version of the manuscript. All the changes are highlighted in yellow in the revised version of the manuscript.

Point-by-point response to the reviewers:

Reviewer 1

This review by Chervy and colleagues is presenting a thorough overview of all the current knowledges regarding adherent and invasive Escherichia coli associated with Crohn’s disease. This article nicely brings together all the past findings regarding AIEC pathogenicity, is illustrated by outstanding figures, and finish by important thoughts regarding the future promises hold by targeting AIEC in Crohn’s disease patients. Moreover, in addition to this list of published virulence factors and mechanisms by which AIEC bacteria can promote chronic intestinal inflammation, the authors also present some interesting open questions and hypothesis throughout the manuscript. I nonetheless have few comments, especially regarding the general organization of the review, as detailed below.

Major comments

- Chapter "2. Virulence factors of the AIEC pathovar" is listing, without giving much details, all known virulence factors which will be subsequently presented in a more detailed manner. Hence, I suggest to delete this chapter that's not bringing anything, expect spoiling the following detailed explanations.

We agree this section was a listing of virulence factors and that it is not necessary to present all of them as we did. For this reason, we removed this part:

“Several AIEC virulence genes, resulting from pathoadaptive mutations, have been identified associated with the AIEC phenotype. Notably, fimH gene coding for the adhesin of type 1 pili that interacts with the host Carcinoembryonic antigen-related cell adhesion molecule 6 (CEACAM6) protein [30–34], chiA coding for chitinase [35] favor the adhesion of the bacteria to the IECs while the genes ibeA coding for an invasin and ompA coding for surface membrane of outer membrane vesicle (OMV) protein promote the invasion of IECs [36,37]. AIEC bacteria are also able to colonize the intestinal mucosa through their interaction with M-cells found at the surface of Peyer’s Patches (PP) by expressing Long Polar Fimbriae (LPF) encoded by the lpf gene, favoring their translocation across the epithelial cells layer and interaction with subepithelial macrophages [38]. AIEC bacteria survive and replicate within macrophages without inducing cell death [39] by expressing gipA gene [40], htrA gene coding for a stress protein [41], dsbA gene coding for a periplasmic oxidoreductase [42] and through the induction of global stress responses [43]. Finally, AIEC bacteria are able to create biofilms at the surface of IECs through the waaWVL operon regulated by the σ(E) pathway in AIEC [44,45]. “

However, we think it is important to present the work concerning the identification of AIEC bacteria and the ongoing clinical trials aiming at identifying molecular biomarkers specific of AIEC carriage in CD patients. Hence, we did not remove the part concerning the diagnostic, unless, the reviewer really thinks this part is not necessary in this review. This chapter is now entitled “The challenging identification of AIEC bacteria in CD patients”.

- Chapter 3 - AIEC in the lumen - is treating a lot about regulation of virulence factors, and hence should be moved down, after all these virulence factors are presented in details. This will greatly improve clarity and avoid unnecessary repetitions.

We thank the reviewer for this pertinent comment. Actually, we decided to present the lifestyle of AIEC bacteria in the lumen at first to follow the chronological lifestyle of the bacteria in the gut (AIEC in the lumen, then adhesion to IECs, then invasion…). However, as we removed the description of the virulence factors in the first part, as suggested by the reviewer, we moved down the part “AIEC in the lumen” to improve clarity. This section is now placed, just before the section “Targeting AIEC”. Accordingly, with these changes, we modified this sentence in the abstract: “In this review, we describe the lifestyle of AIEC bacteria within the intestine, from the interaction with intestinal epithelial and immune cells with an emphasis on environmental and genetic factors favoring their implantation, to their lifestyle in the intestinal lumen.”

- The figures presented are outstanding, and the authors are thank for this contribution. Well done !

Thanks a lot to the reviewer for this nice comment. As the reviewer 3 noticed that the text in the figures was too small, we decided to split the figures to increase their size. The manuscript now contains 4 figures.

- Regarding the use of Bdellovibrio bacteriovorus, a possibility if that this bacterium will broadly target Gram-negative bacteria, which may actually open the niche further for AIEC bacterium (even if AIEC are also targeted). This should be mentioned, as it illustrate the complexity in reversing a dysbiotic state.

This is a very interesting comment that we have now added in the text Page 19 line 8.

“However, it remains necessary to identify the consequences of the presence of B. bacteriovorus on global microbiota composition, especially on Gram-negative bacteria. Indeed, this bacterium could also broadly target Gram-negative bacteria, which may actually open the ecological niche further for AIEC bacterium colonization.”

- Same comment for T6SS - which further highlight the promise of CRIPR approach that can be highly specific to AIEC bacteria.

The following sentence has now been added at the end of the part concerning the T6SS (just before the part concerning CRISPR strategy), Page 19 line 30:

“As specified above, the use of this strategy should rely on the fact that T6SS specifically targets AIEC bacteria, and not all the Gram-negative bacteria in an unspecific manner, to avoid the bloom of AIEC due to the opening of an ecological niche favorable to AIEC growth. This suggests that the development of strategies specifically targeting AIEC should be encouraged, as proposed below.”

Minor comments

- English need proofreading. For example, "the" have to be deleted when plural (Among the microbiota alterations).

This mistake has now been corrected in the abstract. The English has been proofread by a specialized company before submission. As far as we know, the remaining minor mistakes have now been corrected.

- "Another mouse model exhibiting autophagy defects (Tg/eif2ak4-/-) notified the alteration of microbiota composition." While the use of a mouse model can indicate something, it cannot notice anything.

We totally agree with this comment. In Page 15 line 8, the word “notified” has now been changed by “presented” which makes more sense. The new sentence is: “Another mouse model exhibiting autophagy defects (Tg/eif2ak4-/-) presented an alteration of microbiota composition”

- Both figures appeared to be duplicated.

We removed the duplicated figures.

Reviewer 2 Report

The authors presented a very interesting review, in which they summarized the current state of knowledge about virulence factors and the lifestyle of an important for Crohn’s disease patients pathobiont - adherent-invasive E. coli (AIEC). The authors described in detail not only the environmental and genetic factors facilitating the functioning of AIEC, but also indicated potential directions for research on the treatment of this infection.

The minor comments are as follows:

Page 2, line: 16: Please add some references supporting this sentence;” These bacteria were isolated from CD patients in many cohorts and preclinical models of CD suggest an important role of these bacteria in the induction and/or maintenance of intestinal inflammation in CD patients”

Page 2 line 24: Please clarify what do you mean by “attached to the ileum/colon”? Attached to the mucus layer?

Page 2 line 25: “in” before CD is missed

Page 3 line 24: It should be clearly stated that LF28 means (or refers to) one of AIEC strains. At least at the first time.

Page 5 line 11-15: That sentence is unclear. I have a feeling you should remove comma by word ”dysbiosis”.

Page 6 line 12: WT is introduced for the first time. Please explain the abbreviation

Page 8 line 17: HS was previously introduced as “high sugar” but does this expression really refer to a shortcut at this point?

In general, I think the paper is well written. However, authors should avoid using long and complicated sentences.

Author Response

Dear Editor,

We thank the reviewers for their very pertinent comments, which clearly improved the manuscript. We hope we properly answered to their concerns in the revised version of the manuscript. All the changes are highlighted in yellow in the revised version of the manuscript.

Point-by-point response to the reviewers:

The authors presented a very interesting review, in which they summarized the current state of knowledge about virulence factors and the lifestyle of an important for Crohn’s disease patients pathobiont - adherent-invasive E. coli (AIEC). The authors described in detail not only the environmental and genetic factors facilitating the functioning of AIEC, but also indicated potential directions for research on the treatment of this infection.

The minor comments are as follows:

Page 2, line: 16: Please add some references supporting this sentence;” These bacteria were isolated from CD patients in many cohorts and preclinical models of CD suggest an important role of these bacteria in the induction and/or maintenance of intestinal inflammation in CD patients”

This sentence has now been changed by “Preclinical models of CD suggest an important role of these bacteria in the induction and/or maintenance of intestinal inflammation in CD patients”, Page 2 line 13. References supporting these observations have been added (Ref n° 29-31).

Page 2 line 24: Please clarify what do you mean by “attached to the ileum/colon”? Attached to the mucus layer?

“Attached” is not the most appropriate word. We now replaced this word by “associated to”, Page 2 line 22.

Page 2 line 25: “in” before CD is missed

This has now been added, Page 2 line 23.

Page 3 line 24: It should be clearly stated that LF28 means (or refers to) one of AIEC strains. At least at the first time.

This precision has been added Page 3 line 11. “Interestingly, a study using IECs lines revealed that AIEC strain LF82 (one of the AIEC reference strains isolated from CD patients).

Page 5 line 11-15: That sentence is unclear. I have a feeling you should remove comma by word ”dysbiosis”.

For more clarity, this sentence has now been reformulated as follow (Page 13 line 1):

“In genetically susceptible CEABAC10 mice overexpressing human CEACAMs molecules, mimicking CD susceptibility, a HF/HS diet (used as a western diet) induced dysbiosis with a distinctive enrichment in E. coli population. Moreover, this diet increased intestinal permeability, decreased mucus layer thickness, and decreased SCFA concentration and SCFA receptor GPR43 expression.”

Page 6 line 12: WT is introduced for the first time. Please explain the abbreviation

The abbreviation is now explained Page 14 line 45.

Page 8 line 17: HS was previously introduced as “high sugar” but does this expression really refer to a shortcut at this point?

Here, “HS” refers to the commensal strain named HS. It is a widely used strain in laboratories as a control commensal strain. It does not refer to High Sugar in the context of this sentence.

For more clarity, we added E. coli in the sentence Page 3 line 42: “A different regulation of fliC gene expression was noticed in response to intestinal microenvironment components between AIEC bacteria and commensal E. coli HS strain”

In general, I think the paper is well written. However, authors should avoid using long and complicated sentences.

We thanks the reviewer for the pertinent comments and modifications they suggested. We hope we answered most of his concerns.

Reviewer 3 Report

An exclusive review on Adhesive invasive E.coli in Crohn Disease is a very well written, well conceived paper. All the sections are well organised. The only major problem I have is with the 2 Figures. Both the Figures are extremely busy. My only advice is to make it further simplified and of high quality and resolution. It is hard to read some of the sections. Also, both the Figures are placed twice in the manuscript. I think it may be a mistake in organising the pages. Additionally, I would like to see an exclusive section on the future directions in research into AIEC in IBD. 

Overall, an excellent paper -kudos to the authors for a well written review.

Author Response

Dear Editor,

We thank the reviewers for their very pertinent comments, which clearly improved the manuscript. We hope we properly answered to their concerns in the revised version of the manuscript. All the changes are highlighted in yellow in the revised version of the manuscript.

Point-by-point response to the reviewers:

Reviewer 3

An exclusive review on Adhesive invasive E.coli in Crohn Disease is a very well written, well conceived paper. All the sections are well organised. The only major problem I have is with the 2 Figures. Both the Figures are extremely busy. My only advice is to make it further simplified and of high quality and resolution. It is hard to read some of the sections.

To simplify and improve the reading, we separated each figure creating 4 distinct figures. Hence the size of each figures has been increased.

Also, both the Figures are placed twice in the manuscript. I think it may be a mistake in organising the pages.

We removed the duplicated figures.

Additionally, I would like to see an exclusive section on the future directions in research into AIEC in IBD. 

We thank the reviewer for this interesting comment. The following paragraph have now been added Page 22 line 8 and the conclusion has been removed to avoid unnecessary repeats:

“8. Future directions

Many studies have clearly shown the association between the abnormal prevalence of AIEC bacteria in CD patients and the onset of the disease, mostly in industrialized countries. In contrast, very few studies highlighted the presence of these pathobiont bacteria in “in-development” countries such as in South East Asia and Middle East countries, despite the huge increase in the incidence of IBD during the last 10 years. It seems important to evaluate whether AIEC bacteria are common actors in CD all around the world.

One important point in the study of AIEC is the phenotypical characterization of these bacteria. To date, this characterization is not well standardized in all the labs around the world. An effort should be done to standardize the experiments aiming at characterizing these strains to obtain robust prevalence test of AIEC in CD population. The phenotypical characterization is a long process and requires time-consuming experiments before concluding whether the isolated bacteria are AIEC or not. Hence, it seems necessary to develop new tool for a simplified and standardized identification of AIEC bacteria. Instead of studying the bacteria itself, AIEC-specific biomarkers expressed by host cells could be identified (which is one of the aim of the MOBIDIC clinical trial). As an example, immune-dominant antigens expressed by AIEC, leading to an important humoral response, could be identified and could help the development of serological assay predictive of the carriage of AIEC in CD patients. In the case a humoral response specific to AIEC bacteria exists, future researches could focus on the development of vaccinal therapies to prevent AIEC overgrowth.

Finally, preclinical models of CD confirmed the causal or contributing role of AIEC bacteria to the development of CD. However, showing that specifically removing AIEC from intestinal mucosa (as performed in many ongoing clinical trials described above) in CD patients limits the symptoms and relapses of the disease is necessary to validate the role of these bacteria in CD and to propose AIEC targeting as a relevant therapy in CD. To conclude, efforts to better understand the physiology of these bacteria and to elucidate their exact role in CD need to be pursued. It will not only improve diagnostic approaches but will also offer new therapeutic arsenal to propose personal therapies to CD patients colonized by pathobiont AIEC bacteria.”      

Overall, an excellent paper -kudos to the authors for a well written review.

Thanks a lot to the reviewer for this nice comment.